# Globular Aggregates Stemming from the Self-Assembly of an Amphiphilic *N*-Annulated Perylene Bisimide in Aqueous Media

**DOI:** 10.3390/nano11061457

**Published:** 2021-05-31

**Authors:** Manuel A. Martínez, Elisa E. Greciano, Jorge Cuéllar, José M. Valpuesta, Luis Sánchez

**Affiliations:** 1Departamento de Química Orgánica, Facultad de Ciencias Químicas, Universidad Complutense de Madrid, 28040 Madrid, Spain; manuma08@ucm.es (M.A.M.); eegreciano@ucm.es (E.E.G.); 2Centro Nacional de Biotecnología, Campus de la Universidad Autónoma de Madrid, 28049 Madrid, Spain; jcuellar@cnb.csic.es (J.C.); jmv@cnb.csic.es (J.M.V.)

**Keywords:** amphiphiles, hydrophobic effect, perylene bisimides, π-stacking, self-assembly

## Abstract

Herein, we describe the synthesis of highly emissive amphiphilic *N*-annulated PBI **1** decorated with oligo ethylene glycol (OEG) side chains. These polar side chains allow the straightforward solubility of **1** in solvents of different polarity such as water, iPrOH, dioxane, or chloroform. Compound **1** self-assembles in aqueous media by π-stacking of the aromatic units and van der Waals interactions, favored by the hydrophobic effect. The hypo- and hypsochromic effect observed in the UV-Vis spectra of **1** in water in comparison to chloroform is diagnostic of H-type aggregation. Solvent denaturation experiments allow deriving the free Gibbs energy for the self-assembly process in aqueous media and the factor *m* that is indicative of the influence exerted by a good solvent in the stability of the final aggregates. The ability of compound **1** to self-assemble in water yields globular aggregates that have been visualized by TEM imaging.

## 1. Introduction

The key role of water in regulating and biasing a plethora of natural processes has prompted its use in synthetic and supramolecular chemistry [1]. The ability of water to create a dynamic network of H-bonded molecules, a direct consequence of the charge distribution along the molecules, allows the efficient solvation of polar molecules and generates the hydrophobic effect that results in a special interest in constructing supramolecular structures. In these supramolecular aggregates, the balance between hydrophobic and hydrophilic interactions plays a major role in conditioning both the stability and the morphology of such structures [2]. At the same time, the hydrophobic effect provides robustness to the final ensembles that prompts their potential applicability in the area of functional supramolecular materials and, more specifically, in the so-called supramolecular “aquamaterials” in which water molecules favor the efficient self-assembly of small molecules that can be disassembled upon applying specific stimuli [3,4,5]. Among the vast number of supramolecular “aquamaterials”, hydrogels [6] or organic nanocrystals [7] are at the forefront of the field. At the same time, and thanks to the versatility provided by organic synthesis, a large number of examples of supramolecular structures with different morphology and able to exert a variety functions have been reported. Thus, the self-assembly of oligopeptides or amphiphilic benzenetricarboxmides (BTAs) allows the achievement of fibrillar aggregates that can be utilized in biomedical applications [8,9,10], and self-assembled capsules have been reported to act as nanoreactors [9]. Kinetically controlled aggregation processes in aqueous media have been also described for organometallic complexes [11,12,13] or for well-known dyes such as BODIPY [14,15] that yield different species, evolving with time to afford new aggregates with different functions.

Importantly, the decoration of a number of π-conjugated systems with oligo ethylene glycol (OEG) side chains has been thoroughly utilized to generate self-assembling monomeric units suitable to produce aggregates with morphologies of high-aspect ratio [16,17]. Thus, oligo(phenylene ethynylene)s, [18,19] oligo-*p*-phenylenes, [20,21] or hexabenzocoronenes [22] bearing these non-ionic, polar side chains have been reported to generate vesicles, toroidal, or tubular structures in aqueous media. Among all these π-conjugated moieties utilized as scaffolds for the synthesis of non-ionic amphiphiles, perylene bisimides (PBIs) are playing a remarkable role due to their optical and electronic characteristics as well as their trend of self-assembling in an organized manner [23]. Thus, several studies have demonstrated that the self-assembly of amphiphilic PBIs endowed with OEG side chains can be either entropically or enthalpically controlled, depending on the distance from the polar side chains to the central aromatic core [24,25,26].

Despite the examples of non-ionic organic amphiphiles described as forming supramolecular aggregates in aqueous media, a clear structure–function relationship is not accurately established, and new examples of this kind of self-assembling unit are required to expand the knowledge about the organized aggregation of discrete dye molecules in water. Taking into account this background, we have designed and synthesized a new *N*-annulated PBI decorated with two dendritic polar OEG chains attached to the imide functional group and also a single triethylene glycol chain attached to the central nitrogen atom (compound **1** in Figure 1a,b). This amphiphilic *N*-annulated PBI is readily soluble in a variety of solvents and straightforwardly self-assembles in aqueous media. The self-assembly of **1** in water is accompanied by a drastic quenching of the emissive features of the aromatic moiety (Figure 1c) and yields globular aggregates, as the corresponding transmission electron microscopy (TEM) images demonstrate (Figure 1d). A detailed investigation of the energetics of the process has been carried out by applying a denaturation experiment by using water and dioxane, the latter being the denaturant agent. The results presented in this manuscript expand the repertoire of non-ionic organic amphiphiles able to efficiently form organized aggregates in aqueous media.

## 2. Materials and Methods

### 2.1. General

All solvents were dried according to standard procedures. Reagents were used as purchased. All air-sensitive reactions were carried out under argon atmosphere. Flash chromatography was performed using silica gel (Merck, Kieselgel 60, 230–240 mesh or Scharlau 60, 230–240 mesh). Analytical thin layer chromatography (TLC) was performed using aluminium-coated Merck Kieselgel 60 F254 plates (Merck, Darmstadt, Germany). NMR spectra were recorded on a Bruker Avance 300 (Bruker, MA, USA) (^1^H: 300 MHz; ^13^C: 75 MHz) and on a Bruker Avance 700 (^1^H: 700 MHz; ^13^C: 175 MHz) spectrometer using partially deuterated solvents as internal standards. Coupling constants (*J*) are denoted in Hz and chemical shifts (δ) in ppm. Multiplicities are denoted as follows: s = singlet, d = doublet, t = triplet, q = quadruplet, quin = quintuplet, m = multiplet, br = broad. FT-IR spectra in film were recorded on a Bruker Tensor 27 (Bruker, MA, USA) (ATR device) spectrometer. UV-Vis spectra were registered on a Jasco-V630 spectrophotometer (JASCO, Tokyo, Japan) equipped with a Peltier thermoelectric temperature controller. The spectra were recorded in the continuous mode between 200 and 800 nm, with a wavelength increment of 1 nm, a response time of 4 s, and a bandwidth of 1 nm, by using a quartz cuvette (Hellma). Thermal experiments were performed at constant heating rates of 1 °C min^−1^ from 10 to 95 °C in water. Emission spectra were registered on a Jasco FP-6300 (JASCO, Tokyo, Japan) spectrofluorometer. TEM was performed in a JEOL JEM1400 (JEOL, Tokio, Japan) electron microscope operated at 120 kV. Images were directly recorded using a OneView GATAN camera. Samples for TEM imaging were negatively stained with 1% uranyl acetate on carbon grids previously glow-discharged for 15″. Matrix Assisted Laser Desorption Ionization (coupled to a Time-Of-Flight analyzer) experiments (MALDI-TOF) were recorded on a Bruker REFLEX spectrometer.

### 2.2. Synthetic Details and Characterization

For a detailed description of the structural characterization, see Appendix A.

Compounds **2** [27,28] and **6–10** [29] were prepared according to previously reported synthetic procedures and showed identical spectroscopic properties to those reported therein.


**Tetradodecyl 1-(2-(2-(2-Methoxyethoxy)ethoxy)ethyl)-1*H*-phenanthro[1,10,9,8-*cdefg*]ca-rbazole-3,4,9,10-tetracarboxylate (9)**


NaH (25 mg, 0.61 mmol) was slowly added to a mixture of compound **7** (500 mg, 0.45 mmol) and compound **8** (200 mg, 0.61 mmol) in dry THF (11 mL). The resulting mixture was refluxed overnight under argon atmosphere. After that, water (3 mL) was slowly added, and the resulting mixture stirred at 0 °C for 30 min. After evaporation of the solvent under reduced pressure, the residue was extracted with diethyl ether and washed three times with water. The resulting organic layer was dried over MgSO_4_, and the solvent was evaporated under reduced pressure. The obtained residue was precipitated and washed with methanol to afford compound **9** as an orange solid (360 mg, 64%). ^1^H NMR (CDCl_3_, 300 MHz) δ 8.46 (2H, H_1_, d, *J* = 8.0 Hz), 8.24 (2H, H_3_, s), 8.16 (2H, H_2_, d, *J* = 8.0 Hz), 4.84 (2H, H_m_, t, *J* = 5.0 Hz), 4.42 (8H, H_a_, q, *J* = 6.9 Hz), 3.91 (2H, H_n_, t, *J* = 5.0 Hz), 3.39 (4H, H_o+p_, m), 3.29 (2H, H_q_, m), 3.15 (2H, H_r_, m), 3.12 (3H, H_s_, s), 1.86 (8H, H_b_, quin, *J* = 6.9 Hz), 1.52−1.21 (72H, H_c-k_, br), 0.87 (12H, H_l_, t, *J* = 6.9 Hz); ^13^C NMR (CDCl_3_, 75 MHz) δ 169.6, 169.2, 132.1, 131.8, 129.1, 128.0, 127.7, 124.2, 123.3, 121.8, 118.4, 117.9, 77.4, 71.8, 71.2, 71.1, 70.7, 70.6, 65.9, 65.8, 58.9, 46.2, 32.1, 29.9, 29.8, 29.8, 29.6, 29.6, 29.5, 28.9, 26.3, 22.8, 14.3; FTIR (neat) 667, 718, 750, 802, 837, 889, 994, 1023, 1047, 1061, 1129, 1151, 1185, 1201, 1261, 1309, 1348, 1378, 1423, 1469, 1490, 1558, 1587, 1670, 1723, 2850, 2916, 2953 cm^−1^; HRMS (MALDI-TOF) calcd. for C_79_H_121_NO_11_ [M], 1259.8940; found, 1259.8912.


**5-(2-(2-(2-Methoxyethoxy)ethoxy)ethyl)-1*H*-pyrano[3’,4’,5’:4,5]naphtho[2,1,8-*cde*]pyran-o[3’,4’,5’:4,5]-naphtho[8,1,2-*ghi*]isoindole-1,3,7,9(5*H*)-tetraone (10)**


A mixture of compound **10** (0.36 g, 0.29 mmol) and *p*-toluene sulphonic acid monohydrate (0.27 g, 1.43 mmol) in toluene (5 mL) was refluxed overnight. After cooling the mixture at room temperature, the mixture was filtered and washed with methanol to afford compound **10** as a dark red solid (180 mg, quant.). FTIR (neat) 624, 699, 728, 742, 784, 800, 854, 918, 945, 1020, 1059, 1077, 1127, 1180, 1229, 1279, 1328, 1352, 1378, 1397, 1425, 1458, 1483, 1544, 1558, 1601, 1740, 1776, 2853, 2922 cm^−1^; HRMS (MALDI-TOF) calcd. for C_31_H_21_NO_9_ [M+H]^+^, 552.1295; found, 552.1264.


**2,8-bis(13-(2,5,8,11-tetraoxadodecyl)-2,5,8,11-tetraoxatetradecan-14-yl)-5-(2-(2-(2-metho- xyethoxy)-ethoxy)ethyl)-1*H*-pyrido[3’,4’,5’:4,5]naphtho[2,1,8-*cde*]pyrido[3’,4’,5’:4,5]nap- htho[8,1,2-*ghi*]isoindole-1,3,7,9(2*H*,5*H*,8*H*)-tetraone (1)**


A mixture of anhydride **10** (63 mg, 0.114 mmol), amine **6** (107 mg, 0.269 mmol), zinc acetate (84 mg, 0.456 mmol), and imidazole (3 g) was subjected to a few argon/vacuum cycles and heated up at 110 °C overnight. After cooling, the reaction mixture was extracted with chloroform and washed with HCl 1N and saturated sodium chloride solution. The organic layer was dried over MgSO_4_, and the solvent was evaporated under reduced pressure. The resulting residue was purified by column chromatography (silica gel, chloroform/methanol 2%), affording compound **1** as a red solid (69 mg, 46%). ^1^H NMR (CDCl_3_, 300 MHz) δ 8.95 (2H, H_3_, s), 8.81 (2H, H_1_, d, *J*=8.0 Hz), 8.72 (2H, H_2_, d, *J* = 8.0 Hz), 5.04 (2H, H_k_, t, *J* = 4.5 Hz), 4.42 (4H, H_a_, d, *J* = 6.9 Hz), 4.11 (2H, H_l_, t, *J* = 4.5 Hz), 3.67−3.47 (64H, H_(c-i)+(m-p)_, m), 3.33 (12H, H_j_, s), 3.20 (3H, H_q_, s), 0.85 (2H, H_b_, quin, *J* = 6.0 Hz); ^13^C NMR (CDCl_3_, 75 MHz) δ 165.3, 164.2, 135.3, 132.9, 127.7, 124.6, 123.9, 122.2, 121.8, 121.7, 119.7, 119.6, 77.6, 77.2, 72.0, 71.9, 71.1, 71.0, 70.8, 70.7, 70.6, 59.1, 59.0, 47.3, 43.3, 41.0, 38.9; FTIR (neat) 631, 741, 804, 849, 877, 936, 999, 1039, 1105, 1199, 1222, 1251, 1305, 1325, 1352, 1375, 1400, 1427, 1444, 1521, 1558, 1602, 1653, 1689, 2868 cm^−1^; HRMS (MALDI-TOF) calcd. for C_67_H_95_N_3_O_23_ [M], 1309.6356; found, 1309.6368.

## 3. Results

### 3.1. Synthesis of the Amphiphilic N-annulated PBI **1**

The synthesis of the target *N*-annulated PBI **1** was accomplished by following a multistep synthetic protocol by applying reported methodologies for both the OEG side chains and the central aromatic unit [27,28,29,30]. The synthesis of the dendritic wedges started with the two-fold nucleophilic substitution of the two chlorine atoms of commercially available 3-chloro-2-chloromethyl-1-propene by the alkoxy anions generated in situ from triethylene glycol monomethyl ether. The resulting olefine was reacted with borane to generate the corresponding anti-Markovnikov alcohol **2** that is converted into the corresponding amine **6** in a synthetic sequence involving the activation of the hydroxyl group of **3**, nucleophilic substitution with phthalimide potassium salt, and nucleophilic addition of hydrazine to yield amine **6** (Scheme 1) [29]. On the other hand, the *N*-annulated tetraester perylene **7** was prepared by following the previously reported procedure by Achalkumar and coworkers starting from commercial perylene-3,4,9,10-tetracarboxylic dianhydride [27,28]. The *N*-alkylation of **7**, the subsequent conversion to the dianhydride **10** by acidic treatment with *p*-toluenesulphonic acid, and the final nucleophilic addition of amine **6** in the presence of Zn(AcO)_2_ and imidazole affords the amphiphilic *N*-annulated PBI **1** (Scheme 1).

The chemical structure of all new compounds has been confirmed by standard spectroscopic techniques. Regarding compound **1**, it is worth mentioning some relevant resonances observed in the ^1^H NMR spectrum in CDCl_3_. Thus, this spectrum shows a singlet and two duplets corresponding to the aromatic protons of the perylene moiety and triplets at δ ~ 5.1 and 4.4 ascribable to the methylene groups attached to the nitrogen atoms at the aromatic core and the imide groups, respectively. In addition, all the methylene groups corresponding to the OEG side chains are also identified (see Figure 2a and Appendix A).

### 3.2. Self-Assembly of N-annulated PBI **1**

To investigate the self-assembly of amphiphilic **1**, we initially utilized a good solvent like chloroform that favors the efficient solvation of the molecules, and we registered ^1^H NMR spectra at different concentrations. The concentration-dependent experiments in CDCl_3_ show the slight upfield shift of all the three aromatic resonances and a minute shield of the triplet corresponding to the methylene attached to the central nitrogen atom upon increasing the concentration (Figure 2a). However, the resonances corresponding to the methylenes attached to the imide groups (δ ~ 4.4) and those ascribable to the OEG protons experience no shift upon raising the concentration. These shifts, especially those observed for the aromatic resonances, indicate that a weak π-stacking of the aromatic moieties in a rotated fashion could be operative (Figure 2a) [30,31,32]. In addition, we have registered a rotating-frame Overhauser effect spectroscopy (ROESY) NMR experiment in a concentrated solution of **1** (20 mM, CDCl_3_, 25 °C) to extract more information on the self-assembly of this *N*-annulated PBI. The ROESY experiment in CDCl_3_ shows no through-space coupling signals between any of the aromatic resonances. Furthermore, through-space coupling signals are clearly observed between the OEG side chains and these aromatic resonances (Figure 2b). These findings suggest that chloroform could yield small aggregates in which all the pyrrolic units are arranged in a parallel fashion, with the OEG side chains coating the final aromatic units (Figure 1b) [30]. Importantly, the concentrated solutions of **1** in chloroform display a strong orange emission clearly observable by the naked eye (Figure 1c).

After investigating the self-assembly of **1** in a good solvent like chloroform, we studied the influence of the polarity of the solvent on the formation of organized aggregates from amphiphilic *N*-annulated PBI **1**. Thus, we registered the corresponding UV-Vis spectra of diluted solutions of **1** (total concentration, *c_T_* = 10 μM) in different solvents. In chloroform and dioxane, the absorption pattern of these diluted solutions shows two intense bands at λ ~ 525 and 495 nm and a shoulder at λ ~ 465 nm (Figure 3a). This absorption pattern is characteristic of referable molecularly dissolved PBI-based derivatives [23,30]. The solvatochromism observed in the UV-Vis spectra of **1** in these two good solvents is accounted for by considering the donor–acceptor character of the system [33]. Increasing the polarity of the solvent (iPrOH and water) provokes a clear hypso- and hypochromic effect, with UV-Vis spectra showing absorption maxima at λ ~ 543 and 503 nm and a shoulder at λ ~ 478 nm (Figure 3a). These changes are diagnostic of the formation of H-type aggregates in which the molecular units of **1** are stacked one on top of the next one in a face-to-face fashion [34]. The formation of the H-type aggregates of amphiphilic **1** has been corroborated by the strong quenching of the emission intensity observed in water. Thus, the intense emission bands of **1** observed in dioxane, with maxima at λ ~ 596 and 552 nm and the shoulder at λ ~ 649 nm, are significantly reduced upon the addition of increasing amounts of water and almost completely quenched in pristine water (Figure 3b,c).

Taking into account the significance of achieving organized aggregates of the amphiphilic *N*-annulated PBI **1** in aqueous media and the results observed in the emission studies, we have further investigated the thermodynamics of the process in this solvent. Initially, we have performed variable temperature (VT) ^1^H NMR experiments in D_2_O as solvent. At room temperature, the ^1^H NMR spectrum of **1** in D_2_O shows very broad bands for all the resonances diagnostic of a strong supramolecular interaction of the self-assembling units. Increasing the temperature produces a deshielding of all the resonances, and especially the aromatic ones, but also a sharpening effect diagnostic of the efficient interaction of the monomeric units of **1** by the π-stacking of the aromatic moieties and the van der Waals interactions between the OEG side chains (Figure 4a) [35,36]. In fact, diffusion ordered spectroscopy (DOSY) NMR experiments performed in CDCl_3_ at *c_T_* = 20 mM and in D_2_O at *c_T_* = 1 mM (Figure 2c,4c, respectively) afford different values for log (D/m^2^ s^−1^). Thus, whilst in CDCl_3_ this value is −8.19, in D_2_O it is −9.43, revealing the larger size of the aggregates in D_2_O despite the lower concentration utilized. To derive the thermodynamic parameters associated with the self-assembly of **1** in aqueous media, we registered UV-Vis spectra of **1** in water at diluted conditions and at different temperatures. Increasing the temperature to 95 °C results in a minute shift of the absorption maxima that implies a negligible disassembly of the aggregated species of **1** in water by raising the temperature (Figure 4b). Importantly, heating the aqueous solution of **1** does not generate either a scattering effect or turbid dispersions, since the lower critical solution temperature (LCST) is not reached [37,38].

The VT-UV-Vis studies indicate the strong stability of the aggregates formed by compound **1** in aqueous media. Therefore, the thermodynamic parameters associated with the self-assembly of **1** in aqueous media have been accomplished by carrying out a denaturation experiment utilizing diluted solutions of equal concentration in water and dioxane. Mixing these solutions of **1** results in a mixture with invariable total concentration. Thus, the changes observed in the corresponding UV-Vis spectra can be fitted to the solvent-denaturation model (SD) reported by de Greef, Meijer, and coworkers [39]. The addition of the solution of **1** in dioxane to the aqueous solution provokes the gradual disassembly of the aggregates, as demonstrated by the appearance of the absorption maxima at λ ~ 525 and 495 nm and the shoulder at λ ~ 465 nm, diagnostic of the achievement of the molecularly dissolved state (Figure 5a). Plotting the variation of the degree of aggregation (*α*) versus the molar fraction of the good solvent, dioxane, affords a sigmoidal curve diagnostic of an isodesmic mechanism in which all the supramolecular equilibria are defined by a single binding constant *K* (Figure 5b) [40]. Fitting this sigmoidal curve with the SD model provides a value for the Gibbs free energy of monomer association (Δ*G*) of −38.1 ± 4 kJ/mol, a degree of cooperativity *σ* = 1, and the parameter *m* = 19.4 ± 7. Parameter *m* relates the ability of the good solvent to solvate the monomer and, consequently, destabilize the self-assembled structure. Applying the equation Δ*G′* = −*RTLnK*, it is possible to derive a value for the binding constant of *K* = 4.7 × 10^5^ M^−1^.

Finally, we have preliminary registered TEM images that show the aggregates formed by the amphiphilic *N*-annulated PBI **1** in water. Thus, drop-casting an aqueous solution of **1** at *c_T_* = 100 μM onto carbon-coated copper grids shows the formation of globular aggregates with an average diameter of 250 nm (Figure 5c).

## 4. Discussion

The highly emissive *N*-annulated PBI **1** readily self-assembles in aqueous media, giving rise to H-type aggregates, as demonstrated by the corresponding hypo- and hypsochromic effect observed in the UV-Vis spectra in polar solvents and especially in water. The formation of the H-type aggregates is also confirmed by the strong quenching of the orange emission of the monomeric units of **1** upon the addition of increasing amounts of water. Interestingly, heating the diluted aqueous solution of **1** to 95 °C does not result in turbid dispersion due to the release of water molecules interacting by the formation of H-bonds with the polar ethylene glycol side chains. The preliminary results extracted from the VT-UV-Vis experiments, displaying a slight increase in the absorption intensity, suggest the operation of an enthalpically driven self-assembly unlike some reports on the self-assembly of referable amphiphilic PBI-based derivatives [24,25,26]. Some additional studies are ongoing to accurately unravel the thermodynamics of the self-assembly mechanism of compound **1**.

Noteworthily, the self-assembly of amphiphilic **1** yields globular aggregates with an average diameter of 250 nm. Considering the geometry and the dimensions of this amphiphilic self-assembling unit (0.5 and 1.4 nm for the long and short axes of the aromatic core and 3 nm for the longest molecular length by considering a stretched distribution of the OEG side chains, Figure 1b), the globular aggregates should present a hollow nature delimitated by a bilayer of molecules of **1**. The hydrophobic effect generated by the aqueous media strongly favors the π-stacking of the aromatic moieties. A plausible arrangement of the units of **1** to avoid the unfavorable interaction of the hydrophobic aromatic cores with the hydrophilic aqueous media could imply the antiparallel interaction of the self-assembling units, leaving the central OEG side chain pointing inwards and outwards of the aqueous environment. This antiparallel arrangement of the units of **1** contrasts with the parallel distribution of **1** in chloroform (Figure 2b) or some other referable *N*-annulated perylenes [30,31,32]. Unfortunately, our attempts to demonstrate this antiparallel arrangement by performing ROESY experiments in water have been unsuccessful due to the strong supramolecular interactions between the monomeric units of **1** in this solvent. These strong interactions produce broad resonances that prevent an accurate unveiling of clear through-space contacts. Work is currently underway to further investigate the self-assembly of compound **1** in different experimental conditions (concentration and/or solvent) that allows achieving different morphologies for the resulting aggregates. Moreover, additional microscopic techniques are currently underway to further elucidate the supramolecular arrangement of the molecular units of compound **1** in the final aggregates. Importantly, the hollow nature of the aggregates formed by compound **1** in aqueous media could be useful to be applied as nanocontainers able to release the cargo by a controlled disassembly. The disassembly process can be easily followed by an increase in the fluorescence intensity ascribable to the molecularly dissolved state of compound **1**.

## 5. Conclusions

In this manuscript, we report the synthesis of the amphiphilic *N*-annulated PBI **1** decorated with OEG side chains to favor its solubility in a number of apolar and polar solvents and, very especially, in water. The self-assembly of compound **1** has been initially investigated in a good solvent like chloroform and its self-assembling features compared with those exhibited by this amphiphile in water. Compound **1** readily forms H-type aggregates in an isodesmic manner, as demonstrated by the corresponding UV-Vis and fluorescence experiments. The application of the SD model allows deriving the free Gibbs energy for the self-assembly process in aqueous media and the factor *m* that is indicative of the influence exerted by a good solvent in the stability of the final aggregate. The ability of compound **1** to self-assemble in water yields globular aggregates that have been visualized by TEM imaging.

## Data Availability

Not applicable.

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
