# Peer review of "Globular Aggregates Stemming from the Self-Assembly of an Amphiphilic N-Annulated Perylene Bisimide in Aqueous Media"

_nanomaterials, 2021, doi:10.3390/nano11061457_

Round 1

Reviewer 1 Report

The study proposes a synthetic pathway of highly emissive amphiphilic N-annulated perylene bisimide decorated with oligo(ethylene glycol) side chains. These polar side chains allow the compound self-assembly in aqueous media by p-stacking of the aromatic units and van der Waals interactions, favored by the hydrophobic effect. The preliminary study carried out here is certainly interesting and well conducted but it remains very classical. The influence of the hydrophobic effect is not clearly proven. As outlighted in the introduction, studies showing a clear understanding of the property-structure relationships are needed but here the spatial organization of the compound in water has not been sufficiently studied for conclusions. As suggested by the authors themselves at the end of the paper, it would be necessary to carry out a complete study of self-organization in aqueous solution which remains preliminary in this paper. In addition, some of the figures containing multiple spectra are too small to be easily readable (e.g. Figures 2 and 3). In particular in Figure 3, the composition of the mixture shown in the photo c is not clear. 

Author Response

We sincerely appreciate the referee’s remark that considers our manuscript “certainly interesting” and suitable to be published in this Nanomaterials. We are aware that the studies presented in the manuscript are preliminary and that’s why this piece of work is presented as communication. New studies are currently underway to further investigate the self-assembly of the reported amphiphilic N-annulated PBI in order to determine the relevance of the hydrophobic effect on the aggregation of this self-assembling unit and, at the same time, to demonstrate whether or not the self-assembly of this system is enthalpically or entropically controlled. In addition, new experiments, especially cryo-TEM, have to be performed to elucidate the organization of the molecules in the aggregated state.

Regarding the remark raised by the referee concerning the scholarly presentation of Figures 2 and 3, we have modified these figures accordingly in order to improve their quality.

Reviewer 2 Report

Dear authors, I enjoyed reading and reviewing your manuscript.

I have some minor revisions to suggest for the publication of this paper.

-Page 4, line 155, there appears EOG instead of OEG, I suppose that it is due to a typing mistake, but authors must correct it.

-Page 5, line 188, the same mistake is repeated.

-Page 8, line 307, the same mistake is repeated.

-Page 5, line 172, substitute "The chemical structure of all new compounds have been confirmed by the habitual spectroscopic techniques." by "The chemical structure of all new compounds has been confirmed by standard spectroscopic techniques."

-Page 5.- line 177, remove at in the following sentence "..and at the imide groups, respectively."

-Page 7, line 248, when the authors describe the UV-Vis spectra of compound 1 results, they referred in the manuscript to figure 4b, but it must be referred to figure 4c.

-Finnaly, authors should include some references, related to influence of ethylene glycols upon self-assembly of amphiphiles (DOI: 10.1021/la500403v, DOI:10.1007/s00396-009-2122-0, DOI: 10.1201/9781420089608.ch4), and influence of solvents on self-assembled nanomaterials (DOI: 10.1039/C9TC00889F, DOI: 10.1021/acsomega.7b00049, DOI: 10.1002/smll.201803563).

Author Response

We thank the reviewer for raising this remark. Following his/her suggestions, we have amended the mistakes highlighted by the referee. Regarding the suggested references, we have included them as references 4,5 16 and 17 in the revised version of the manuscript.

Reviewer 3 Report

In this paper, the authors describe the synthesis and self-assembly of an amphiphilic perylene bisimide derivative. The self-assembling behavior in water was investigated in an appropriate manner. NMR and vt-spectroscopic studies afforded reasonable insights into the physicochemical aspect of amphiphilic self-assembly in water, which is also discussed carefully. I don’t have any concern in recommending the present contribution for publication in this journal.

Author Response

We appreciate so much this statement raised by the referee.

Round 2

Reviewer 1 Report

The readability of figures 2 and 3 has clearly been improved. However, the authors do no clearly respond to my request to explain in what their perylene bisimide was different from the previous one and will help to elucidate the structure-function relationship about non-ionic organic amphiphiles compared to the previous compounds. It would have been clearer to precise it at the end of introduction before the description of the compound solubility.